# Representation Learning for Spatial Multimodal Data Integration with Optimal Transport

**Xinhao Liu**
Department of Computer Science
Princeton University
Princeton, NJ 08544
xl5434@princeton.edu

**Benjamin J. Raphael**
Department of Computer Science
Princeton University
Princeton, NJ 08544
braphael@princeton.edu

## Abstract

Spatial sequencing technologies have advanced rapidly in the past few years, and recently multiple modalities of cells – including mRNA expression, chromatin state, and other molecular modalities – can be measured with corresponding spatial location in tissue slices. To facilitate scientific discoveries from spatial multi-omics sequencing experiments, methods for integrating multimodal spatial data are critically needed. Here we define the problem of spatial multimodal integration as integrating multiple modalities from related tissue slices into a Common Coordinate Framework (CCF) and learning biological meaningful representations for each spatial location in the CCF. We introduce a novel machine learning framework combining optimal transport and variational autoencoders to solve the spatial multimodal integration problem. Our method outperforms existing single-cell multi-omics integration methods that ignore spatial information. Our method allows researchers to analyze tissues comprehensively by integrating knowledge from spatial slices of multiple modalities.

## 1 Introduction

Spatial sequencing technologies are advancing rapidly in recent years, particularly spatially resolved transcriptomics technologies that measure mRNA expression at thousands of locations in a tissue slice jointly with their spatial coordinates. Spatially resolved transcriptomics was named the "Method of the Year" by *Nature Methods* in 2020 [Marx, 2021] and has been applied in dozens of tissues and diseases include cancer [Thrane et al., 2018, Ji et al., 2020] and Alzheimer's disease [Chen et al., 2020]. Recently, spatial sequencing has been extended to profile other molecular modalities including chromatin accessibility [Deng et al., 2022], protein abundances [Liu et al., 2020], and epigenetic modifications [Lu et al., 2022]. Each of these modalities describes a unique aspect of the underlying biological state of the cells at each spatial location. To improve the definition of cell types and cell states, and to comprehensively characterize spatial heterogeneity of tissues, it is advantageous to integrating experimental data from multiple sources and modalities into a consensus representation. An integrated view of experimental data has already proven to facilitate further biomedical discoveries, such as cancer therapeutic target discovery [Yao et al., 2023, Liang et al., 2023].

In silico alignment and integration for *single-cell* sequencing data are mostly concerned with learning a common low dimensional latent space to embed the measured feature vectors of cells from different modalities. However, the challenge with multi-modal integration of spatial sequencing data is that cells in spatial sequencing data already occupy a common low dimensional space, namely the coordinate system of the physical tissue slice they are sequenced from. Thus, in contrast to single-cell integration tasks, the integration of spatial data not only needs to embed each cell's features into a common space, but also needs to find an embedding that respects the physical space.

NeurIPS 2023 AI for Science Workshop.

We define the task of spatial multimodal integration as transferring data onto a shared reference called a Common Coordinate Framework (CCF) [Rood et al., 2019]. Andersson et al. [2021] uses a similar definition for integrating transcriptomics data. The reference CCF is a set of points with fixed coordinates in a physical space, usually the tissue from which the spatial sequencing data are taken. Spatial multimodal sequencing of the tissue involves taking multiple slices from the same region, then for each slice, a different omics layer is profiled spatially. Therefore, the task of integration is to register each spot from each slice onto the CCF. Additionally, we want to learn a representation for each spot on the CCF such that this embedding captures information from its corresponding spots in all omics layers and defines the true underlying biological state (Fig. 1). This learned representation integrates knowledge of all omics measurements and opens the door to multiple downstream tasks, such as highly accurate cell type classification or translation and imputation between modalities.

In this work, we present a machine learning framework based on optimal transport (OT) and variational autoencoders (VAE) for spatial multimodal data integration. Optimal transport is gaining popularity in the machine learning community for its ability to explore geometric information in data, which makes it a suitable mathematical tool for spatial modeling [Cang and Nie, 2020]. On the other hand, multimodal variational autoencoders have been successfully applied to single-cell multimodal integration [Wu and Goodman, 2018]. In this work, we combine the geometric awareness of OT and the multimodal representation learning ability of VAE to arrive at an effective and efficient algorithm for the spatial multimodal data integration task. We show the effectiveness of our method through four real multi-omics spatial sequencing datasets, on which our method achieves state-of-the-art performance. To the best of our knowledge, this is the first method designed for spatial multimodal integration, and we anticipate our method to be useful for building tissue atlases across both normal and diseased tissues, such as in the Human BioMolecular Atlas Program, the Human Tumor Atlas Network, and other international spatial sequencing projects. Our code is publicaly available at `https://github.com/x-h-liu/otvi`.

## 1.1 Related works

Integration is not a new problem for scRNA-seq data. Packages such as Seurat [Stuart et al., 2019] have become standard in the field now and scRNA-seq integration software are now being used as a standard step in single cell data preprocessing pipelines. More recently, researchers introduced single-cell multimodal integration as single-cell multi-omics sequencing technologies become available. Due to the unique obstacles presented by multi-omics data, especially distinct and often unpaired feature spaces, machine learning and AI techniques are useful in this field. scAI [Jin et al., 2020] integrates paired scRNA-seq and scATAC-seq data through an iterative matrix factorization approach. scDART [Zhang et al., 2022] designs a neural network to translate between modalities. GLUE [Cao and Gao, 2022] aligns single cell data from any modality based on variational autoencoders and prior knowledge of regulatory interactions. These methods either work with paired data (ground-truth correspondence between cells from multiple data sets is known), or assume some prior knowledge in modality translation. The only unsupervised integration methods are OT-based methods SCOT [Demetci et al., 2022] and Pamona [Cao et al., 2022], but these methods are typically used on small-scale data sets. Moreover, all the aforementioned methods are designed for single-cell data, while spatial sequencing data present further challenges.

Another line of work focuses on the integration of multiple spatially resolved transcriptomics datasets. PASTE [Zeira et al., 2022] and PASTE2 [Liu et al., 2023] use OT to align two adjacent spatial slices. STUtility [Bergenstråhle et al., 2020] aligns 10x Genomics Visium slices using image registration. GPSA [Jones et al., 2023] and eggplant [Andersson et al., 2021] both create a CCF for spatial slices and learn a Gaussian Process to register slices onto the CCF. However, although these methods are designed for spatial sequencing, they only integrate transcriptomics data and do not handle other modalities. To the best of our knowledge, our method is the first unsupervised method to integrate spatial multi-omics slices without the assumption of any prior knowledge.

## 2 Background: Optimal Transport

**Notation** For a topological space $X$, let $\mathcal{P}(X)$ be the set of probability measures on $X$. Let $\delta$ be the Dirac function. Let $\mathbf{1}_n$ denote the $n$-dimensional vector of ones, and let $\langle \cdot, \cdot \rangle_F$ denotes the Frobenius dot product. Let $\Sigma_n = \{p \in \mathbb{N}_n^+; \sum_i p_i = 1\}$ be simplex of histograms with $n$ bins.

## 2.1 Optimal Transport

Optimal transport (OT) addresses the problem of comparing and transporting between probability measures. One pays a cost to transport one probability measure into another, and OT is interested in finding the most cost-effective transformation. Consider probability measures $\mu \in \mathcal{P}(X), \nu \in \mathcal{P}(Y)$ on spaces $X$ and $Y$ respectively. Assume we are given a cost function $c : X \times Y \to [0, +\infty]$ where $c(x, y)$ measures the cost of transporting one unit of probability mass from $x \in X$ to $y \in Y$. The Kantorovich formulation [Kantorovich, 1942] of optimal transport seeks a joint measure $\pi \in \mathcal{P}(X \times Y)$ minimizing

$$\inf_{\pi} \int_{X \times Y} c(x, y) d\pi(x, y) \tag{1}$$

subject to the constraints

$$\pi(A \times Y) = \mu(A) \text{ and } \pi(X \times B) = \nu(B) \tag{2}$$

for all measurable sets $A \subseteq X, B \subseteq Y$. Let $\Pi(\mu, \nu)$ be the set of all joint measures $\pi$ that satisfy the above constraints, called the set of *transport plans* between $\mu$ and $\nu$. When $\mu$ and $\nu$ are discrete measures, $\mu = \sum_{i=1}^{n} u_i \delta_{\mathbf{x}_i}$ and $\nu = \sum_{j=1}^{m} v_j \delta_{\mathbf{y}_i}$, where $\{\mathbf{x}\}_{i=1}^{n}$, $\{\mathbf{y}\}_{j=1}^{m}$ are discrete samples. Let $\mathbf{u} \in \Sigma_n, \mathbf{v} \in \Sigma_m$, then the set of transport plans becomes a polytope $\Pi(\mu, \nu) = \{\pi \in \mathbb{R}^{n \times m} | \pi \mathbf{1}_m = \mathbf{u}, \pi^T \mathbf{1}_n = \mathbf{v}\}$. The Kantorovich problem is equivalent to the linear program

$$\min_{\pi \in \Pi(\mu, \nu)} \langle C, \pi \rangle_F, \tag{3}$$

where $C \in \mathbb{R}^{n \times m}$ is a matrix defined by $C_{ij} = c(\mathbf{x}_i, \mathbf{y}_j)$.

## 2.2 Fused Gromov-Wasserstein Optimal Transport

An important aspect of the Kantorovich OT is that the transport cost matrix, $C$, reflects the metric of transport distance between two domains $X$ and $Y$. This becomes a limiting factor in cases where there is underlying structure *within* each domain, since this structural information is not included in the transport cost. To address this issue, structured optimal transport [Alvarez-Melis et al., 2018, Peyré et al., 2016] has been proposed to compare structural information between domains. Peyré et al. [2016] proposed the Gromov-Wasserstein OT distance to compare two distributions even if they do not lie in the same ground space. When the two domains to compare come from the same underlying metric-measure space, Fused Gromov-Wasserstein (FGW) distance [Titouan et al., 2019] has been proposed to utilize both inter-domain *feature* information and intra-domain *structure* information in the same framework. Specifically, let $d(\mathbf{x}_i, \mathbf{x}_j)$ be the metric within domain $X$, $d'(\mathbf{y}_i, \mathbf{y}_j)$ be the metric within domain $Y$, and $c(\mathbf{x}_i, \mathbf{y}_j)$ be the metric across domains as defined above, the FGW-OT aims to find a transport plan that minimizes

$$\min_{\pi \in \Pi(\mu, \nu)} (1 - \alpha) \sum_{i,j} c(\mathbf{x}_i, \mathbf{y}_j) \pi_{ij} + \alpha \sum_{i,j,k,l} (d(\mathbf{x}_i, \mathbf{x}_k) - d'(\mathbf{y}_j, \mathbf{y}_l))^2 \pi_{ij} \pi_{kl}. \tag{4}$$

The FGW optimal transport problem is a non-convex quadratic program and can be solved using the Frank-Wolfe algorithm [Frank et al., 1956, Levitin and Polyak, 1966].

## 3 Methods

Suppose we are given $K$ spatial slices from the same tissue, each with a different omics measurement (Fig. 1). Each omics measurement has a distinct feature set, denoted as $\mathcal{V}_k$ for the $k$-th measurement. For example, for spatial transcriptomics (RNA-seq), $\mathcal{V}$ is the set of genes while for spatial ATAC-seq $\mathcal{V}$ is the set of chromatin regions. Therefore, the $k$-th spatial omics experiment yields a pair $(X^k, Z^k)$, where $X^k \in \mathbb{N}^{n_k \times |\mathcal{V}_k|}$ are the feature measurements of each modality, $Z^k \in \mathbb{R}^{n_k \times 2}$ are the 2D coordinates of each spot, and $n_k$ is the number of spots on the $k$-th slice. Row $\mathbf{x}_i^k \in \mathbb{N}^{|\mathcal{V}_k|}$ of $X^k$ is the feature vector of spot $i$ in the feature space of omics measurement $k$. Row vector $\mathbf{z}_i^k \in \mathbb{R}^2$

gives the $(x, y)$ coordinate of spot $i$ on the $k$-th slice. Following [Zeira et al., 2022], we encode the spatial location of each spot in a pairwise Euclidean distance matrix $D^k \in \mathbb{R}_+^{n_k \times n_k}$, where $d_{ij}^k$ is the Euclidean distance between spot $i$ and spot $j$ on the slice calculated from $Z^k$. Therefore, the $k$-th spatial slice with omics feature space $\mathcal{V}_k$ is represented by a tuple $(X^k, D^k)$.

We assume we are given $Z^* \in \mathbb{R}^{n_* \times 2}$, the coordinates of spots on the center slice, or Common Coordinate Framework (CCF), where $n_*$ is the number of spots of the CCF. We similarly encode $Z^*$ as $D^*$ as described above.

Our goal is to learn a representation matrix $X^* \in \mathbb{R}^{n_* \times m}$ of the CCF, where $m$ is the dimension of the low-dimensional embedding of each spot on the CCF. Additionally, we want to learn $K$ probabilistic mappings that map spots from each slice $k$ to the CCF. Each mapping $\pi^k$ is a matrix $\pi^k = [\pi_{ij}^k] \in \mathbb{R}^{n_k \times n_*}$, where $\pi_{ij}^k$ measures the probability that spot $i$ on slice $k$ is mapped to spot $j$ on the CCF (Fig. 1).

Given $(X^1, D^1) \dots (X^K, D^K)$, and $D^*$, we aim to find a representation matrix $X^*$, and mappings $\pi^1 \dots \pi^K$ that minimizes the following cost function

$$\sum_{k=1}^{K} \left[ (1-\alpha) \sum_{i,j} c\left(X_{i.}^k, X_{j.}^*\right) \pi_{ij}^k + \alpha \sum_{i,j,k,l} \left(D_{ik}^k - D_{jl}^*\right) \pi_{ij}^k \pi_{kl}^k \right] \tag{5}$$

subject to the regularity constraints that each of $\pi^1, \dots, \pi^K$ are probabilistic couplings:

$$\pi^1 \mathbf{1}_{n_*} = \mathbf{a^1}, (\pi^1)^T \mathbf{1}_{n_1} = \mathbf{c}, \dots, \pi^K \mathbf{1}_{n_*} = \mathbf{a^K}, (\pi^K)^T \mathbf{1}_{n_K} = \mathbf{c}, \tag{6}$$

where $\mathbf{a^1} \dots \mathbf{a^K}$ and $\mathbf{c}$ are uniform distributions over spots in slice 1 through $K$, and the center slice, respectively. Notice the similarity of these constraints with Eq. 2. $c(\cdot, \cdot)$ is a feature dissimilarity function which computes the dissmilarity between the two feature vectors. Note that since the feature vectors for the center slice are low dimensional, the two input vectors to $c(\cdot, \cdot)$ are of different dimensions. $\alpha$ is a hyper-parameter balancing the contribution between the two terms to the objective function. We solve this problem using a Block Coordinate Descent algorithm that alternates between optimizing the representation matrix $X^*$ given the current mappings $\pi^1, \dots, \pi^K$ and optimizing the mappings $\pi^1, \dots, \pi^K$ given the current embedding $X^*$ (Fig. 1).

### 3.1 Optimization of representation matrix $X^*$ for fixed alignments $\pi^1, \dots, \pi^K$

Given fixed mappings $\pi^1 \dots \pi^K$, the optimization problem (5) becomes

$$\min_{X^*} \quad \sum_{k=1}^{K} \sum_{i,j} c(X_{i.}^k, X_{j.}^*) \pi_{ij}^k \tag{7}$$

That is, given the correspondence between spots in the CCF and spots in each omics slice, we are interested in a low dimensional feature representation $X^*$ of each spot in the CCF that encodes the biological information of all its corresponding spots in all omics layers.

First, note that $\widetilde{X^k} = n_* \cdot (\pi^k)^T \cdot X^k \in \mathbb{R}^{n_* \times |V_k|}$ is the transferred feature matrix of the $k$-th modality onto the center slice. After computing $\widetilde{X^k}$ for all modalities, each spot in the center slice now receives $K$ feature vectors, one for each modality. Inspired by MultiVI [Ashuach et al., 2023], for each spot in the CCF, we model the observed feature vectors as generated by a low-dimensional latent variable $\mathbf{z} \in \mathbb{R}^m$ representing the common cell state underlying all omics observations. To infer $\mathbf{z}$ from $\mathbf{x}^1$ through $\mathbf{x}^K$ for a specific spot, we introduce $K$ encoders for variational posterior $\mathbf{z}$, each denoted as $q_k(\mathbf{z}|\mathbf{x}^k, \theta)$, and $K$ decoders for generative distributions $\mathbf{x}$, each denoted as $p_k(\mathbf{x}^k|\mathbf{z}, \theta)$. With prior distribution $p(\mathbf{z})$, we train the VAE by optimizing the following evidence lower bound:

$$\mathbb{E}_{\mathbf{z} \sim \prod_k q_k(\mathbf{z}|\mathbf{x}^k)} \left[ \log \left( \prod_k p_k(\mathbf{x}^k|\mathbf{z}) \right) \right] - \text{KL} \left( \prod_k q_k(\mathbf{z}|\mathbf{x}^k) \Big\| p(\mathbf{z}) \right). \tag{8}$$

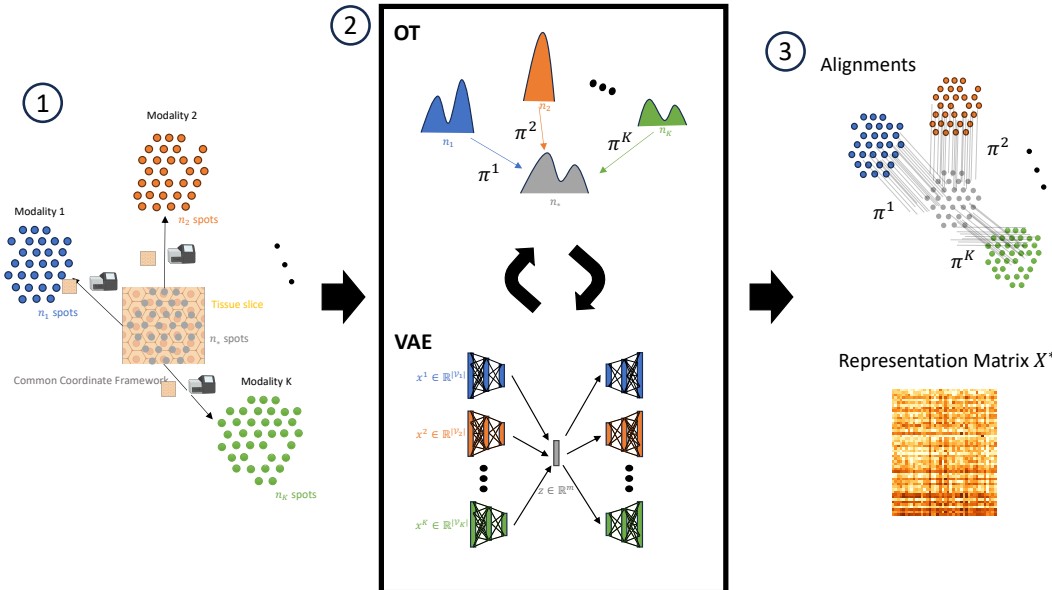

**Figure 1:** Overview of our spatial multimodal integration framework. Given multiple slices of different modalities of the same tissue and a CCF of the region (step 1), our method alternates between OT alignment learning and VAE representation learning (step 2) to compute mappings between each omics slice and the CCF, as well as a low-dimensional representation of each spot on the CCF (step 3).

Briefly, $K$ encoders, one for each modality, map their respective feature vectors onto $K$ low dimensional embeddings, respectively. The final embedding $\mathbf{z}$ is taken as the average of the $K$ embeddings. Then, $K$ decoders map the same $\mathbf{z}$ to reconstruct $K$ feature vectors respectively. The overall loss of the network is the sum of the $K$ reconstruction errors and a penalty to minimize the distance between the $K$ embeddings. In practice we use the implementation of MultiVI [Ashuach et al., 2023]. The details of the training procedure are in Appendix A.

### 3.2  Optimization of alignments $\pi^1 \dots \pi^K$ for fixed representation matrix $X^*$

For fixed representation matrix $X^*$, the optimization problem (5) becomes

$$\min_{\pi^1,\dots,\pi^K} \quad \sum_{k=1}^{K} \left[ (1-\alpha) \sum_{i,j} c\left(X_{i\cdot}^k, X_{j\cdot}^*\right) \pi_{ij}^k + \alpha \sum_{i,j,k,l} \left(D_{ik}^k - D_{jl}^*\right) \pi_{ij}^k \pi_{kl}^k \right] \tag{9}$$

$$s.t. \quad \pi^1 \mathbf{1}_{n_*} = \mathbf{a^1}, (\pi^1)^T \mathbf{1}_{n_1} = \mathbf{c}, \dots, \pi^K \mathbf{1}_{n_*} = \mathbf{a^K}, (\pi^K)^T \mathbf{1}_{n_K} = \mathbf{c}. \tag{10}$$

Since there is no term depending on any two $\pi$'s simultaneously, the $K$ variables can be independently optimized. For $\pi^1$,

$$\min_{\pi^1} \quad (1-\alpha) \sum_{i,j} c\left(X_{i\cdot}^1, X_{j\cdot}^*\right) \pi_{ij}^1 + \alpha \sum_{i,j,k,l} \left(D_{ik}^1 - D_{jl}^*\right) \pi_{ij}^1 \pi_{kl}^1 \tag{11}$$

$$s.t. \quad \pi^1 \mathbf{1}_{n_*} = \mathbf{a^1}, (\pi^1)^T \mathbf{1}_{n_1} = \mathbf{c}. \tag{12}$$

The above problem is exactly the same as the formulation of the Fused Gromov-Wasserstein OT problem (Eq. 4). Here, the cross-domain feature distance function $c(X_{i\cdot}^*, X_{j\cdot}^1)$ is defined by running $X_{j\cdot}^1$ through the encoder network $q_1(\mathbf{z}_1|X_{j\cdot}^1)$ trained from the previous iteration and calculating the Euclidean distance between $X_{i\cdot}^*$ and $\mathbf{z}_1$. Since $X_{i\cdot}^*$ and $\mathbf{z}_1$ are of the same dimensionality, this solves the problem of $c(\cdot, \cdot)$ taking in vectors of two different dimensions. The optimization problems for other $\pi^k$'s are defined analogously. We solve each of these problems using the conditional gradient algorithm as implemented in the POT library [Flamary et al., 2021].

We altrenate the above two steps for a fixed number of iterations. Appendix Algorithm 1 gives the pseudo-code for our training procedure.

## 4 Results

### 4.1 Datasets

To test the performance of our method on spatial multimodal data integration, we used four pairs of spatial multi-omics slices, each with a different pair of modalities, from two recently published technologies for spatial epigenome-transcriptome co-sequencing [Zhang et al., 2023]. The two technologies are spatial-ATAC-RNA-seq, which simultaneously profiles chromatin accessibility and mRNA expression, and spatial-CUT&TAG-RNA-seq (spatial assay of cleavage under targets and tagmentation and RNA using sequencing), which simultaneously profiles histone modifications and mRNA expression. The authors applied both spatial-ATAC-RNA-seq and spatial-CUT&TAG-RNA-seq to P22 mouse brain coronal sections and generated four spatial co-profiling slices: RNA with ATAC, RNA with H3K27me3 histone modification, RNA with H3K27ac histone modification, and RNA with H3K4me3 histone modification. Each slice has about 9,000 spots. For each spot on each slice, both a RNA profile and an epigenome profile are simultaneously sequenced. The dimensionality is over 10,000 for RNA measurements, and over 100,000 for epigenome measurements.

To evaluate integration methods, we treat each epigenome-transcriptome co-profiling slice as *two* independent slices, one with epigenome profile and one with RNA profile. The ground-truth spot correspondence is known from the experiment which obtained two omics profiles simultaneously for each spot. Therefore, our evaluation metric of the performance of integration methods is whether they can recover this ground-truth spot-spot relationship. Let $G$ be the set of spot pairs $(i, j)$ from the two slices that are ground-truth correspondence. We define the *Accuracy* of an alignment to be

$$Acc(\pi) = \sum_{(i,j) \in G} \pi_{ij}, \tag{13}$$

where $\pi$ is an alignment matrix from one modality slice to another (For our method, $\pi = \pi^2 \circ \pi^1 = \pi^1 \cdot (\pi^2)^T$). $Acc(\pi) \in [0, 1]$ is the sum of elements $i, j$ in $\pi$ for pairs $i, j$ that are in the set of ground truth correspondence, and can be interpreted as the percentage of ground-truth correspondences recovered by the method. This metric may be too strict for some applications since high accuracy requires aligning many spots to their ground-truth corresponding spots, while in practice many methods only aim to align spots to spots of the same type. Thus, we also evaluate performance using the Label Transfer Adjusted Rand Index (LTARI) (Appendix B) as proposed in [Liu et al., 2023].

We use the spot coordinates of the co-profiling slice as the spot coordinates of the CCF. We set the embedding dimension $m = 16$, and initialize the two alignment matrices $\pi^1, \pi^2$ randomly. We then alternate between the two block coordinate steps for 30 iterations. We run the experiments on an Nvidia Tesla P100 GPU. For integrating two slices with about 9,000 spots each, each iteration spends about 400 seconds on learning the OT alignment matrix, and about 1000 seconds on learning the multimodal VAE.

### 4.2 Baselines

We compared our method with two state-of-the-art unsupervised single-cell multi-omics integration methods, SCOT [Demetci et al., 2022] and Pamona [Cao et al., 2022]. SCOT integrates two single-cell datasets from two modalities by first building a $k$-nearest-neighbor (kNN) graphs of cells within each modality based on the feature vectors of each cell. Then it builds the distance matrices of cells within each modality by taking the distance from one cell to another as the length of the shortest path in the kNN graph, which can be shown to approximate the geodesic distance on the embedded manifold [Tenenbaum et al., 2000]. After computing two intra-modality geodesic distance matrices, SCOT uses Gromov-Wasserstein optimal transport [Peyré et al., 2016] to compute an alignment matrix $\pi^{scot} \in \mathbb{R}^{n_1 \times n_2}$ between cells from each modality. In addition to the alignment, SCOT also projects cells from one modality to the feature space of the other modality using barycentric projection as a way of embedding cells from both modalities into one common space.

**Table 1:** Results on integration of four multimodal spatial slice pairs

| Dataset | Method | Acc | LTARI(RNA) | LTARI(Epigenome) | FOSCTTM |
|---|---|---|---|---|---|
| RNA-ATAC | Ours | **0.89** | **0.91** | **0.91** | N/A |
| | SCOT | 0.0002 | 0.043 | 0.033 | **0.37** |
| | Pamona | 0.0002 | 0.026 | 0.030 | 0.38 |
| RNA-H3K4me3 | Ours | **0.83** | **0.86** | **0.84** | N/A |
| | SCOT | 0.0001 | 0.099 | 0.060 | 0.35 |
| | Pamona | 0.0003 | 0.068 | 0.043 | 0.35 |
| RNA-H3K27ac | Ours | **0.92** | **0.93** | **0.93** | N/A |
| | SCOT | 0.0002 | 0.12 | 0.094 | 0.34 |
| | Pamona | 0.0003 | 0.051 | 0.040 | **0.30** |
| RNA-H3K27me3 | Ours | **0.83** | **0.86** | **0.82** | N/A |
| | SCOT | 0.0002 | 0.077 | 0.077 | 0.36 |
| | Pamona | 0.0002 | 0.040 | 0.037 | **0.32** |

Pamona follows the same philosophy as SCOT, but uses partial optimal transport [Caffarelli and McCann, 2010] and embeds cells into a common space using Laplacian eigenmap [Belkin and Niyogi, 2003].

### 4.3 Evaluations

We ran SCOT, Pamona, and our method on all four spatial multimodal pairs and computed accuracy, RNA LTARI, and epigenomic LTARI (Table. 1). Our method achieves very high performance on all four pairs. The accuracy, RNA LTARI, and epigenomic LTARI are higher than 80% for all four pairs, meaning that our method is able to recover more than 80% of ground-truth relationships between spots from spatial slices of two different modalities. On the contrary, the accuracy measure of both SCOT and Pamona are very low compared to our method, meaning that both methods are not able to recover the exact relationship between spots.

The low accuracy of SCOT and Pamona is because the *accuracy* defined above measures the exact one-to-one relationship between spots. However, for single-cell integration methods such as SCOT and Pamona, the focus is to embed all cells in a common space such that batch effects are eliminated and cells are clustered by cell types instead of experimental artifacts. These methods do not need to recover the exact one-to-one correspondence between spots in order to find a good embedding. However, the fact that our method is able to strictly recover this information is impressive and demonstrates our method's ability to faithfully recover the ground-truth biological relationship between spatial multimodal slices.

To demonstrate that SCOT and Pamona did not catastrophically fail for these datasets and are reasonable baseline methods, we computed fraction of samples closer than the true match (FOS-CTTM)(Appendix B), a metric used to evaluate these methods in the original publications. FOSCTTM measures the closeness between ground-truth correspondence pairs in the joint embedding space calculated by the integration method, and is the metric that both SCOT and Pamona are targeting. FOSCTTM cannot be computed for our method because we only compute an embedding for spots in the CCF, not all cells from all slices. The FOSCTTM for both methods are around 0.3 on all four datasets. This number is consistent with the trend reported in the SCOT paper (Fig. 2C of Demetci et al. [2022]), where in the simulation study the FOSCTTM of SCOT increases from 0.1 to 0.4 (lower is better) as the number of samples in the test dataset exceeds 300. Since the four datasets in this work contain over 9,000 samples each, SCOT and Pamona both scale well to datasets of this large size and infer a good common embedding space. However, this also points out another potential reason for their low accuracy in terms of recovering the exact one-to-one relationship: the methods were previously evaluated on small datasets (number of samples $< 1000$), while the datasets evaluated here are large. On the contrary, our method is able to recover the ground-truth relationship despite the large dataset size.

To evaluate the quality of the center slice inferred by our method, we performed Leiden clustering [Traag et al., 2019] of the 16-dimensional embedding of spots on the four inferred center slices, and

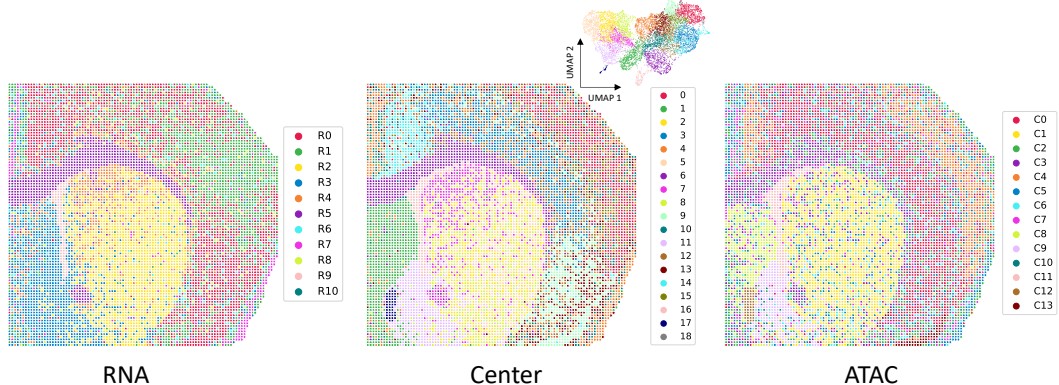

**Figure 2:** Single-modality clusters of the RNA-ATAC slice pair and the clusters of the inferred CCF slice.

compared the resulting clusters with the clusters inferred on single-modalities in the original study [Zhang et al., 2023]. For the RNA-ATAC pair, we find that by integrating information from both modalities into a center slice and computing a joint embedding, the integrative analysis of our method preserves the specificity of each modality while also finds more heterogeneous cell types (Fig. 2). The original study reported 11 clusters in the RNA slice and 14 clusters in the ATAC slice, while we find 19 clusters in the integrative center slice. The 19 clusters preserves both RNA and ATAC-specific clusters. For example, cluster 7 of the center slice reflects cluster R4 of the RNA slice, which is not present in the ATAC slice, while cluster 17 of the center slice reflects cluster C12 of the ATAC slice, which is not present in the RNA slice. In addition to preserving modality-specific cell types, the center slice also reveals underlying heterogeneity not covered by either modality. For example, both the RNA and the ATAC slice reports two clusters in the upper right region (R0 and R1 for RNA, C0 and C4 for ATAC), while the center slice indicates there are three cell types (clusters 0, 3, 9), the location of which are different from information presented in either single-modality slice. This shows that by integrating knowledge from multi-omics spatial slices, our method is able to infer a CCF that captures additional biological heterogeneity within the tissue region and uncovers information hidden in single modality. We observed similar patterns integrating RNA with histone modifications (Fig. S1, S2, S3).

## 5    Conclusion

In this work, we develop a method that uses optimal transport and variational autoencoders to integrate multiple spatial slices of different -omics modalities. We show through four pairs of spatial multi-omics slices that our method accurately aligns spatial spots with distinct modalities, and our inferred representation of the common coordinate framework discovers more cell type heterogeneity than single modality alone. We expect our method to be useful in large-scale cell atlas projects.

## Acknowledgments and Disclosure of Funding

This work was supported by NIH/NCI grant U24CA248453 awarded to B.J.R. We want to thank Julian Gold and Peter Halmos for their helpful feedback on the manuscript.

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

# APPENDIX

## A  Details of the variational autoencoder

For the following discussion we assume the method is integrating two modalities, spatial RNA with spatial ATAC, but the method is easily generalized to multiple slices from multiple omics technologies. The following discussion mirrors the setup of [Ashuach et al., 2023]. For each spot on the center slice, let $\mathbf{x}^1 \in \mathbb{N}^{|\mathcal{V}_1|}$ be the projected RNA feature vector and $\mathbf{x}^2 \in \mathbb{N}^{|\mathcal{V}_2|}$ be the projected ATAC feature vector. The probability of observing $\mathbf{x}_i^1$ counts in gene $i$ is modeled as a negative binomial distribution

$$\mathbf{x}_i^1 \sim \text{NegBin}(\ell\rho_i, \theta_i) \tag{14}$$

where $\ell$ is a per-spot scaling factor that captures spot-specific bias, $\rho_i$ is normalized gene frequency of gene $i$, and $\theta_i$ is the dispersion parameter of gene $i$. The probability of observing $\mathbf{x}_i^2$ in chromatin region $i$ is modeled as a Bernoulli distribution

$$\mathbf{x}_i^2 \sim \text{Ber}(\ell p_i r_i) \tag{15}$$

where $p_i$ is the true accessibility of region $i$ and $r_i$ models region-specific bias. The decoder network $p_1(\mathbf{x}^1|\mathbf{z})$ learns the normalized gene frequency vector $\rho$ from the latent variable $\mathbf{z}$ with the reconstruction loss as the likelihood function of negative binomial distribution. The decoder network $p_2(\mathbf{x}^2|\mathbf{z})$ learns the true region accessibility vector $p$ from the latent variable $\mathbf{z}$ with the reconstruction loss as the likelihood function of Bernoulli distribution. $\ell, \theta_i, r_i$ are optimized directly.

Both the encoder networks $q_1(\mathbf{z}_1|\mathbf{x}^1), q_2(\mathbf{z}_2|\mathbf{x}^2)$ learn their own latent representation $\mathbf{z}_1, \mathbf{z}_2$, and the prior distributions for both are isotropic multivariate normal distribution with mean 0 and variance 1. Then, the latent variable $\mathbf{z}$ of the spot representing the underlying cell state is inferred as the average $\mathbf{z} = \frac{1}{\mathbf{z}_1 + \mathbf{z}_2}$. Finally, the loss function has a term that encourages $\mathbf{z}_1$ and $\mathbf{z}_2$ to be close to each other by minimizing the symmetric KL divergence between them. The overall loss function of the network is

$$L = L_R + L_A + \text{symmKL}(q_1, q_2) \tag{16}$$

where $L_R$ is the RNA reconstruction error and $L_A$ is the ATAC reconstruction error. Since the experiments done in this work are all integrating RNA sequencing data with ATAC-like sequencing data, we use the default implementation in [Ashuach et al., 2023] of these networks.

## B  Evaluation metrics

**Accuracy**  Since for all the slice pairs used in this work, we know the ground-truth correspondence between spots on one slice and spots on the other slice, the accuracy measure is simply

$$Acc(\pi) = \sum_{(i,j) \in G} \pi_{ij} \tag{17}$$

$\pi$ is an alignment matrix from one modality slice to another (For our method, $\pi = \pi^2 \circ \pi^1 = \pi^1 \cdot (\pi^2)^T$). $Acc(\pi) \in [0, 1]$ is the sum of elements $i, j$ in $\pi$ for pairs $i, j$ that are in the set of ground truth correspondence $G$, and can be interpreted as the percentage of ground-truth correspondences recovered by alignment $\pi$.

**LTARI**  *Accuracy* defined above can be too strict as an alignment only receives credit for spot pairs that it aligns exactly correct. In most alignment tasks, it usually suffices to only align cells to the correct cell *type* on the other slice, rather than the exact cell. Therefore, we also evaluate alignment matrices using Label Transfer Adjusted Rand Index (LATRI) [Liu et al., 2023], which measures the ability of an alignment $\pi$ to transfer labels from one slice to the other. Specifically, for each spot $j$ on one of the slices, the alignment $\pi$ induces a new cell type label for this spot by assigning it $\ell(j) = \ell(\text{argmax}_i \pi_{ij})$, the label of spot $i$ on the other slice that spot $j$ is most likely aligned to according to $\pi$. This assignment transfers the labels of spots from the first slice to the second slice.

We then compare this transferred labeling of spots with the ground truth labeling of the second slice and compute the Adjusted Rand Index (ARI) of the two clusterings, and call the resulting ARI Label Transfer ARI (LTARI). A higher LTARI indicates that $\pi$ better preserves the cell type labeling of the two slices by aligning cells to cells of the same cell type on the other slice.

For the task of multimodal integration, the two slices to align come from two modalities. The RNA slice has a cell type labelling based on RNA clustering, and the epigenomic slice has a cell type labelling based on epigenomic clustering. We computed LTARI for both labelling, termed RNA LTARI and epigenomic LTARI.

**FOSCTTM**  FOSCTTM, or fraction of samples closer than the true match, introduced by MMD-MA [Liu et al., 2019], is a metric defined on the joint embedding space inferred by integration methods. For many single-cell data integration algorithms, instead of finding an alignment mapping between cells from one dataset to cells from the other dataset, they often embed cells from both datasets into a common space. That is, they learn a low-dimensional representation for all cells in both datasets such that cells of the same cell type from both datasets are close together in the common space. Then, for each cell from one dataset in the common space, we compute its Euclidean distance in the common space to all cells from the other dataset. We then compute the fraction of those distances that are smaller than the distance to the cell's ground truth correspondence in the other dataset. The FOSCTTM score is defined as the average of the fractions over all cells. For a perfect integration, all cells would be closest to their true match, yielding a FOSCTTM of zero. Therefore, a lower FOSCTTM score indicates a better integration result.

Since our spatial multi-omics alignment method only produces an alignment between slices, but does not embed all spots into a common space (in contrast, we *do* embed spots of the CCF to a common space), FOSCTTM does not apply to our method. Both SCOT and Pamona computes a common space, so we evaluate FOSCTTM on their alignment results to show that their alignments are valid on our dataset.

---

**Algorithm 1:** Spatial Multi-omics Integration

**Input:** $K$ spatial slices $(X^1, Z^1) \ldots (X^K, Z^K)$, each with a different modality measurement $X$ and coordinate matrix $Z$; Coordinate matrix of the center slice $Z^*$; Dimensionality of the embedding $m$; Number of iterations $N$;

**Output:** Representation matrix $X^* \in \mathbb{R}^{n_* \times m}$ of the center slice; $K$ mappings $\pi^k \in \mathbb{R}^{n_k \times n_*}$

---

1 Encode $Z^1 \ldots Z^K$ as Euclidean distance matrices $D^k \in \mathbb{R}_+^{n_k \times n_k}$ ;
2 Encode $Z^*$ as Euclidean distance matrix $D^* \in \mathbb{R}_+^{n_* \times n_*}$ ;
3 Initialize mappings $\pi^1 \ldots \pi^K$ randomly ;
4 Initialize representation $X^*$ randomly ;
5 **for** *iteration* $i = 1 \ldots N$ **do**
6      Optimize $X^*$, fixing $\pi^1 \ldots \pi^K$ (Eq. (8));
7      Optimize $\pi^1 \ldots \pi^K$, fixing $X^*$ (Eq. (9));
8 Optimize $X^*$, fixing the final $\pi^1 \ldots \pi^K$ (Eq. (8));
9 **return** $X^*, \pi^1 \ldots \pi^K$

---

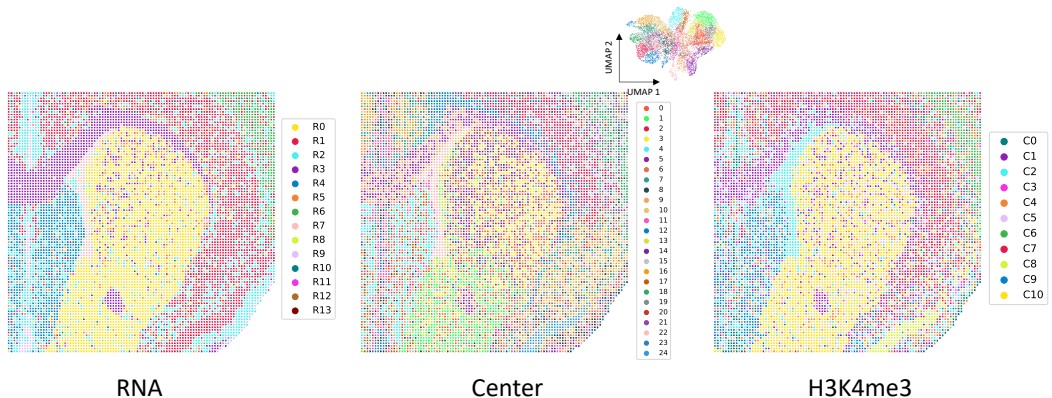

**Figure S1:** Single-modality clusters of the RNA-H3K4me3 slice pair and the clusters of the inferred CCF slice.

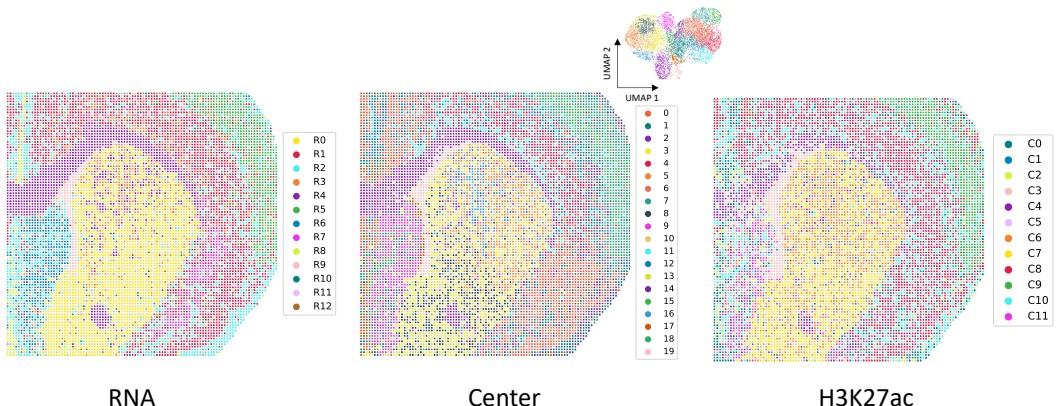

**Figure S2:** Single-modality clusters of the RNA-H3K27ac slice pair and the clusters of the inferred CCF slice.

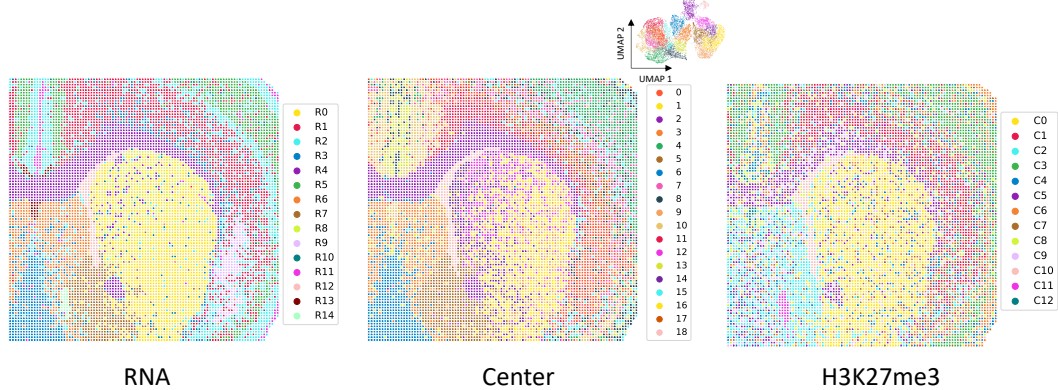

**Figure S3:** Single-modality clusters of the RNA-H3K27me3 slice pair and the clusters of the inferred CCF slice.

