# OpenReview forum: "Representation Learning for Spatial Multimodal Data Integration with Optimal Transport"
_NeurIPS.cc/2023/Workshop/AI4Science — NeurIPS2023-AI4Science Poster_

### Official Review · Reviewer_s8uZ · 2023-10-17
**Advances in Spatial multimodal data integration using RL**

**Rating:** 7
**Confidence:** 3

**Review:**

The article introduces a machine learning framework that combines optimal transport and variational autoencoders to address the spatial multimodal integration problem. This framework has the potential to significantly enhance our understanding of complex biological systems and foster scientific discoveries from spatial multi-omics experiments. The article excels in providing a clear and concise explanation of the technical aspects, enhancing accessibility for a broad readership. However, a potential limitation lies in the absence of an in-depth discussion on the computational requirements and scalability of the proposed framework.

I find the proposed machine learning framework to be a mathematically robust implementation, employing a novel approach to optimize the representation matrix by training the VAE and alignments using the conditional gradient. The authors have elucidated the technical intricacies of the framework; however, I recommend a more detailed training algorithm description, potentially including pseudocode, in the next manuscript version. Furthermore, a refined Figure 1, incorporating precise dimensional and architectural information, would aid readers seeking a deeper understanding of the technicalities.

The integration of multimodal spatial data, encompassing both transcriptomics and epigenomics data, using optimal transport is a particularly innovative aspect of the paper. The presented experimental results are impressive, showing significant improvements over baselines, especially in LTARI, across all datasets. However, a direct comparison of the baseline experiment conditions is essential to confirm the impact of the proposed model. For instance, while Pamona represented LTARI (RNA) higher than 0.2 in the reference (Liu et al. 2023), it is lower than 0.1 in this paper. Such a direct comparison would elucidate the relative performance of the proposed framework in comparison to existing methods.

I suggest that the authors include a simple downstream task, such as cell type classification, to illustrate the necessity of the proposed model effectively. This would provide a tangible example of how integrated multimodal spatial data can enhance the accuracy and comprehensiveness of downstream analyses. Additionally, further applications, such as constructing spatial gene co-expression networks, would illuminate the potential uses of the proposed model, enabling a deeper understanding of the underlying biological mechanisms within tissues.

Nonetheless, a potential limitation of the paper lies in the lack of a detailed discussion regarding the computational requirements and scalability of the framework. This omission could be crucial for researchers interested in applying the model to large-scale datasets. Furthermore, while the experimental results are promising, a more extensive evaluation of the proposed framework across a broader array of datasets and experimental conditions would be beneficial, as mentioned earlier.

In summary, I believe this paper constitutes a valuable contribution to the field of multimodal spatial data integration, with significant implications for real-world biological research.

---

### Meta-Review · Area_Chair_jGzP · 2023-10-27

**Recommendation:** Accept (Poster)
**Confidence:** 5

**Metareview:**

This paper introduces a method for integrating single-cell spatial multi-omics data to derive meaningful representations of spots on the Common Coordinate Framework (CCF). This is achieved by harnessing the geometric awareness of Optimal Transport (OT) through the utilization of a Variational AutoEncoder (VAE) architecture.

This research is highly interesting as it represents a pioneering effort in designing a solution for spatial multi-omics integration,  encapsulating substantial insights into cellular heterogeneity. Notably, the experimental results demonstrate that only the proposed method excels in capturing the one-to-one relationship between spots.

However, as pointed out by the reviewer, a significant limitation arises from the lack of experiments applying the integrated data to other downstream tasks, such as cell type classification or the construction of spatial gene co-expression networks. The absence of such experiments leaves the ambiguity of whether these representations hold biologically meaningful characteristics.

I recommend conducting additional extensive experiments to validate this method further. However, I support the acceptance of this paper, as I concur with the reviewer's argument that this application could have significant implications in this domain.